# Quality Evaluation of Shrimp (*Parapenaeus longirostris*) Treated with Phenolic Extract from Olive Vegetation Water during Shelf-Life, before and after Cooking

**DOI:** 10.3390/foods10092116

**Published:** 2021-09-07

**Authors:** Dino Miraglia, Marta Castrica, Sonia Esposto, Rossana Roila, Roberto Selvaggini, Stefania Urbani, Agnese Taticchi, Beatrice Sordini, Gianluca Veneziani, Maurizio Servili

**Affiliations:** 1Department of Veterinary Medicine, University of Perugia, Via San Costanzo 4, 06126 Perugia, Italy; dino.miraglia@unipg.it (D.M.); rossana.roila@unipg.it (R.R.); roberto.selvaggini@unipg.it (R.S.); 2Department of Health, Animal Science and Food Safety “Carlo Cantoni”, Università degli Studi di Milano, Via Celoria 10, 20133 Milan, Italy; marta.castrica@unimi.it; 3Department of Agricultural, Food and Environmental Sciences, University of Perugia, Via San Costanzo s.n.c., 06126 Perugia, Italy; stefania.urbani@unipg.it (S.U.); agnese.taticchi@unipg.it (A.T.); beatrice.sordini@unipg.it (B.S.); gianluca.veneziani@unipg.it (G.V.); maurizio.servili@unipg.it (M.S.)

**Keywords:** olive vegetation water, phenolic extract, antimicrobial, antioxidant, cooked shrimps, sensory traits

## Abstract

The focus of this study was to assess the quality traits and sensory profile of cooked rose shrimps (*Parapenaeus longirostris*) treated with a phenolic extract, derived from olive vegetation water (PEOVW). To achieve the aim, four different groups of shrimps were analysed, specifically the control (CTRL) group, where the shrimps were soaked in tap water; sulphites (S) group with shrimps soaked in 0.5% sodium metabisulfite tap water solution, phenolic extract (PE) group where a tap water solution containing 2 g/L of phenols was used; and PE+S group where the shrimps were dipped in 0.25% sodium metabisulfite tap water solution containing 1 g/L of phenols. The groups were then stored at 2 °C and analysed on the day of packaging (D0), after 3 (D3), 6 (D6), and 8 (D8) days. On each group, microbiological parameters such as *Enterobacteriaceae*, mesophilic and psychrotrophic bacteria, and colorimetric indices were investigated on six (*n* = 6) shrimps before cooking, while the evolution of the phenolic content, antioxidant activity, and sensory analysis during the storage period were evaluated on cooked shrimps. Regarding colour coordinates, there were no noteworthy variations overtime nor between groups, while it is important to note that the microbiological results for the PE group showed at each time interval and for all the considered parameters, significantly lower values than the other groups (*p* < 0.05). This result is very interesting when considered further in correlation with the sensory analysis, where shrimps mainly in PE and secondarily in PE+S groups were shown to retain the freshness characteristics better than the other groups (α = 0.01), without giving the shrimps any particularly bitter and pungent sensations typical of the olive phenolic compounds. In conclusion, the results obtained in this study give PEOVW the potential to be valorised in the food sector and, above all, it could represent a sustainable solution to reduce the use of synthetic additives.

## 1. Introduction

Crustaceans are fishery products of high commercial value that are widespread all over the world [1,2]. Among these, *Parapenaeus longirostris*, or deep-water pink shrimp represents one of the most consumed and commercialized species in Mediterranean countries and Europe [3,4]. It is very appreciated for its delicate taste, the consistency of the meat, and its high nutritional value. Similar to all crustaceans, however, it is characterized by a very short shelf life and refrigeration alone is not sufficient to preserve its organoleptic characteristics for an adequate amount of time [5]. Shrimps are highly perishable due to their chemical-physical properties (poor collagen, high pH and humidity, non-protein nitrogen), so immediately after death, microbial and endogenous enzymes begin the degradation processes, quickly leading to loss of quality and unacceptability of the product [6,7]. The main problem with fresh shrimps after harvest is the formation of dark pigments called black spot or melanosis, caused by polyphenoloxidase enzyme systems (PPO). In the presence of molecular oxygen, PPO can catalyze phenols oxidation into quinones which subsequently undergo polymerization giving rise to these high molecular weight pigmented compounds, even under refrigerated storage. Melanosis may occur before microbial deterioration and, although it does not appear to be harmful to public health, the commercial value of these products decreases. To avoid or slow down the enzymatic browning during the shrimp marketing, the fishing industry has several strategies including the use of synthetic preservatives. Sulphites, in particular sodium metabisulfite (E223), are the most widely used food additives to prevent melanosis development in crustaceans. However, the presence of preservatives is perceived with a certain mistrust by consumers, due to the adverse effects that some substances may generate in sensitive individuals [8]. For this reason, several studies in recent years, have focused on the use of, plant based natural compounds, that can replace or reduce the number of sulphites, commonly used in inhibiting melanosis in crustaceans [9,10,11,12]. In this regard, bioactive molecules such as phenols derived from fruits and vegetables have shown promising results. Their antibacterial and antioxidant activity [13,14,15,16] seem to delay the development of black spots (melanosis) and preserve the quality of crustaceans during storage [10,11,12]. Notably in a previous study, Miraglia et al. [17] obtained, even with limited effects on melanosis formation, a significant reduction in lipid oxidation, total volatile basic nitrogen, and bacterial counts, using a phenolic extract derived from olive vegetation water (PEOVW) in rose shrimps (*Parapenaeus longirostris*). Olive mill waste (OMWs) vegetation waters are highly polluting, yet their high phenols content [6,18,19] makes them an extremely interesting by-product for functional and food applications [20]. For these reasons, the reuse of oil mill waste water in the food sector represents a valorization strategy in line with the EU objectives of moving towards a circular economy system, reducing the environmental impact and, at the same time, disposal costs [21]. The aim of the present study was to evaluate the quality traits and sensory profile of cooked pink shrimps treated with an olive vegetation water phenolic extract. The sensory evaluation will be able to define the sensory variations occurring in relation to the cold storage period and, at the same time, it will allow us to understand if the different treatments can be perceived and described by the tasters.

## 2. Materials and Methods

### 2.1. Experimental Design Methodology

The phenolic extract used in this study was derived from olive vegetation water as described by Esposto et al. [22] and stored at −20 °C until its use. The extract (containing a concentration in phenolic compounds of 518.4 mg/g), at the moment of its utilization, had the following composition:108.9 mg/g of 3,4-dihydroxyphenylethanol (3,4-DHPEA);18.6 mg/g of p-hydroxyphenylethanol (p-HPEA);30.8 mg/g of verbascoside;360.1 mg/g of the dialdehydic form of elenolic acid linked to 3,4-DHPEA (3,4-DHPEA-EDA).

Freshly caught deep-water rose shrimps (*Parapenaeus longirostris*) were collected, without any treatment, and immediately kept in ice during the transportation to the Food Inspection Laboratory at the Department of Veterinary Medicine, University of Perugia (Italy). Upon arrival, the shrimps were randomly divided into four groups: Control (CTRL), sulphites (S), phenolic extract (PE), and phenolic extract + sulphites (PE+S). Each group was immersed for 15 min at 12 °C in four different solutions (2/1 solutions/shrimp): The CTRL group was soaked in tap water; S in 0.5% sodium metabisulfite tap water solution, PE group in a tap water solution containing 2 g/L of phenols obtained by adding 3.86 g PEOVW/L, and the PE+S group was dipped in 0.25% sodium metabisulfite tap water solution containing 1 g/L of phenols obtained by adding 1.93 g PEOVW/L. The shrimps were then drained and arranged into different polystyrene containers (300 g/container, four per group) and packed in an oxygen-permeable package. Throughout the experimental period, the containers were stored at 2 °C and analysed on the day of packaging (D0), after 3 (D3), 6 (D6), and 8 (D8) days of packaging. At each sampling time, specimens were randomly removed from storage and subjected to physicochemical, microbiological, and sensory analyses. The cooking procedure, when necessary, consisted of immersing the samples into boiling water (1.5 L) and cooking them until the shrimps reached the surface of the boiling water bath [23].

### 2.2. Microbiological Analyses

Analyses to determine changes in the microbiological profile during shelf-life were carried out each time on six (*n* = 6) shrimp samples per group, following the same sampling procedure as reported by Miraglia et al. [17]. Briefly, *Enterobacteriaceae* was enumerated using the Violet Red Bile Glucose agar (VRBG, Oxoid, Basingstoke, UK) and the plates were incubated at 37 °C for 24 h, while Plate Count Agar (PCA, Oxoid) was used to enumerate the mesophilic and psychrotrophic bacteria after incubation at 30 °C for 48 h and 4 °C for 10 days, respectively. All microbiological counts were expressed as Log CFU/g of sample and the analyses were performed in duplicate.

### 2.3. Physico-Chemical Analyses

The extraction of phenolic compounds from 5 g of shrimp samples (without carapaces), was performed according to Miraglia et al. [17] and then, the analysis of the phenolic extracts was conducted through the HPLC-DAD-FLD method according to Selvaggini et al. [24]. The phenolic content was expressed as mg/g sample. The DPPH (2,2-diphenyl-1-picrylhydrazyl) radical-scavenging activity of shrimp samples was determined according to the method described by Brand-Williams et al. [25]. In brief, 200 µL aqueous shrimp extract was added to 3.8 mL of DPPH methanolic solution (25 mg/L). The mixture was then allowed to stand in the dark for 20 min at room temperature. Finally, the absorbance of the mixture was measured using a Cary 100 Bio UV-Visible Spectrophotometer (Varian Analytical Instruments, Walnut Creek, CA, USA) at 515 nm. The results were expressed as μmol of Trolox equivalents (TE)/g of shrimp using the calibration curve of Trolox (0.01–0.1 μmol). The phenolic content and antioxidant activity were assessed at all sampling times on cooked shrimp samples, except for day 0 where the analyses were also carried out on raw samples. Regarding colour, a Minolta Chromameter 400 colorimeter (Minolta Ltd., Osaka, Japan) was used to assess the brightness (L*), redness (a*) and yellowness (b*), of the shrimps. The analyses were performed in triplicate on the cephalothorax carapace.

### 2.4. Sensory Analysis

#### 2.4.1. Triangle Test

To evaluate significant sensory differences among the CTRL, S, PE, and PE+S samples at each phase of storage, a discriminant sensory test was carried out on the cooked shrimps. In this regard, the shrimp groups at each storage time (D0, D3, D6, and D8) were subjected to a triangular test (ISO 4120:2004), comparing, for each type and at different storage times, the CTRL sample with each counterpart (S, PE, and PE+S), and all the other groups in pairs. For this sensory analysis, the samples were prepared by following the procedure reported by Erickson et al. [23]. Briefly, each group of shrimps was immersed in boiling water (1.5 L) and cooked until they rose to the surface. Soon after, the shrimps were placed in an ice bath for 5 min. After draining, the samples were served at room temperature in white plastic plates, each one marked with a three-digit code. The panel of the triangle test consisted of 36 trained tasters (24 female and 12 males aged 25 to 55), according to the significance parameters chosen (Pd = 50%; α = 0.01, ISO 4120:2004). Each panelist was presented with one different and two identical samples. All of the three samples were presented to the panelist at once, and the panelists were instructed to sensory evaluate the samples from left to right. The panelist was instructed to identify the odd sample and record his answer. Samples from the three replicates of the same group were served randomly so that all the replicates of the same group were presented in an equal number of times. Each judge was presented with six different combinations. Significant differences were determined based on the results reported in the significance tables (ISO 4120:2004) and to the chosen values of Pd (50%) and α risk (α = 0.01).

#### 2.4.2. Quantitative-Descriptive (QDA) Test

The quantitative-descriptive analysis (QDA) method (UNI EN ISO 13299:2016) was used and an unstructured, linear graphical scale (100 mm) was converted to numerical values (0 to 10 conventional units). Sensory quality was characterized on the basis of sensory attributes, grouped in the following: Appearance, odor, taste, texture, and overall quality. Descriptors were chosen and defined during the panel discussion and then verified in the preliminary session. This analytical sensory analysis was also performed on cooked CTRL, S, PE, and PE+S shrimps at each storage time (D0, D3, D6, and D8), by 12 trained assessors using a sensory profile sheet. For this purpose, the panelists were trained both for the use of the sensory profile sheet for the correct interpretation of the sensory descriptors previously selected, as well as for the use of the relative unstructured scale, following the procedure reported by Erickson et al. [23]. The cooking of the samples was carried out following the same procedure used for the triangle test [23]. Each taster was subjected to randomized sequences of samples in balanced order (and therefore, different each time) to avoid the fact that the judgments were influenced by the order of tasting. The ratings were made individually. For each group and storage time, the shrimps were placed on plastic plates and coded with a random three-digit number. Results were included in the multivariate statistical analyses PCA and O-PLS-DA.

### 2.5. Statistical Analysis

To determine whether the differences were significant, after the normality verification (Shapiro-Wilk), microbiological and color data were analyzed by the nonparametric test. The Mann-Whitney test for between-group comparisons and the Wilcoxon signed rank test was used for within-group comparisons during storage days. In all cases, a value of *p* < 0.05 was considered to indicate statistical significance. The data were analyzed with the use of SPSS statistical software, version 26.0. The SIMCA 13.0 chemometric package (Umetrics AB, Umeå, Sweden) was used to conduct the principal component analysis (PCA) and orthogonal least squares regression analysis (O-PLS-DA) on the instrumental and sensory data obtained.

## 3. Results

### 3.1. Microbiological Results

The microbial trend in pink shrimps during the storage period at 2 °C is illustrated in Figure 1 A–C. At each time interval and for all the considered parameters, the PE group showed significantly lower values than the other groups (*p* < 0.05). However, even in this group, a significant increase in microbial levels was observed from day 6 compared to the starting values (D0; *p* < 0.05). These findings suggest that the antibacterial action exerted by polyphenols occurs, particularly, during the treatment phase of the shrimps with the phenolic extract and its effectiveness tends to decrease over time. These evidences are in agreement with the quantities of polyphenols detected in the analyzed samples, in which the phenolic fractions recorded at D0 decreased with the passing of storage days. Similar trends were also found in previous studies where the PEOVW, at the highest concentrations, resulted in a reduction in TVC and *Enterobacteriaceae* from the first analysis time [16,17,26]. These results confirm the antimicrobial action of phenolic compounds derived from olive vegetation waters, repeatedly reported in the literature, although their effectiveness depends on the concentration of the bioactive molecules and the microbial species considered [16,17,27,28,29]. From this point of view, the treatment with a solution containing 1 g/L polyphenols and 0.25% sodium metabisulphite tested in this study exerted a significant inhibition only on *Enterobacteriaceae* growth at the end of the storage period (PE+S vs. CTRL: *p* < 0.05), only partially confirming the results obtained in a previous study where PE+S also limited the development of mesophilic and psychrotrophic bacteria [17]. Under the same experimental conditions, the microbial evolution could have been conditioned by intrinsic differences between the two batches of shrimp, such as: The season of the fishing year, the supplier, the undergone manipulations, and the initial hygiene level of the catch.

### 3.2. Physico-Chemical Characteristics

#### 3.2.1. Phenolic Composition of Raw and Cooked Shrimps

The evolution of phenolic composition during the storage period, before (Table 1) and after cooking (Table 2), in shrimps treated with PEOVW was evaluated. The loss of total phenolic content in cooked shrimps (Table 2), compared to raw shrimps (Table 1), was similar between the groups, with average percentage values of 69.3% in PE and 68.5% in PE+S samples, with a peak recorded at the end of the storage period (77.1% and 86.7% in PE and PE+S groups, respectively). This peak coincided with the total disappearance at the end of the storage period of 3,4-DHPEA, which until then had been the most abundant phenolic fraction in both groups (Table 1). Indeed, with cooking, this compound suffered progressive percentage losses, particularly in PE shrimps, although these were partially compensated by the release of 3,4-DHPEA and by hydrolysis of 3,4-DHPEA-EDA [22,30]. In PE+S samples, losses were also progressive but more limited, probably due to a protective action exerted by sulphites during boiling [17] (Table 2). This evidence is in accordance with the concentration of these compounds in the raw product, where hydroxytyrosol recorded the highest recovery rates, particularly in the PE+S samples, while 3,4-DHPEA-EDA progressively decreased, until it was no longer detected on D8 (Table 1), confirming the findings of previous studies [16,17]. Therefore, it is plausible that the absence of 3,4-DHPEA in cooked shrimps at the end of storage (Table 2), has been a direct consequence of the depletion of its precursor in raw shrimps (Table 1). Indeed, 3,4-DHPEA-EDA is an extremely reactive potent antioxidant, which decreases rapidly over time following oxidative processes and hydrolytic phenomena [22,30]. These reactions were further accelerated by the heat treatment, so that boiling completely degraded this compound from the first detection. These results are partially in agreement with those reported by Taticchi et al. [31], where cooking tomato sauce at 100 °C, resulted in losses of the total added phenols, with lower percentages (up to 39.9%). In particular, 3,4-DHPEA-EDA suffered the greatest decrease (up to 54.6%), while 3,4-DHPEA and p-HPEA actually increased in concentration. Conversely, Silva et al. [32], found that baking extra virgin olive oil, in the presence of food, reduced the concentration of all polyphenolic components, with a decrease of the total phenolic content of about two-thirds, and a loss of 3,4-DHPEA-EDA of 98.0%.

#### 3.2.2. Colour Results

The data in the literature on shrimps colour are varied, not only in relation to the species and treatment considered, but also in terms of the evolution of lightness (L*), redness (a*), and yellowness (b*) coordinates over time. In particular, some authors report a decrease in L* values during conservation [11,33,34], whereas others, an increase [3,34]. In this study, lightness constantly increased over time, although with a different intensity between treatments, so that at the last observation the S samples were the lightest (*p* < 0.05), while the groups treated with the phenolic extract had the lowest values (*p* < 0.05) (Figure 2A). Moreover, in a previous study, shrimps treated with PEOVW, alone or in combination with sodium metabisulphite, were the least shiny at the end of shelf life, while the group treated with sulphites only had a tendency to remain lighter [17]. This can be attributed to the ability of sulphites to bleach crustaceans, as reported by Martínez-Alvarez et al. [35]. In regards to the redness index (a*), a decrease was observed throughout storage in all batches, although at D8 a slight increase brought the values similar to the starting ones (Figure 2B). Treatment as well as storage, did not affect the a* coordinate, therefore confirming previous findings [17]. Little change in red intensity over time was also recorded by Senapati et al. [34], in contrast to some authors [11,36,37,38,39] who showed a decrease and others [6,33] who instead reported an increase. Similar to lightness, the yellow index (b*) also increased over time, starting from day 6 in the CTRL and PE groups (*p* < 0.001) and at the last detection in the S and PE+S groups (*p* < 0.001; Figure 2C). The presence of sulphites thus seems to have slowed down the yellowing of shrimps, which were therefore less yellow at the end of the observation period (*p* < 0.05), confirming the trend observed in a previous study [17]. As observed by other authors, crustaceans generally tend to yellow during storage and the antimelanostic agent such as sulphites are able to slow this trend down [6,11,33,34,35,36,37,38,39].

### 3.3. Triangle Test

A discriminant test was conducted to determine the overall sensory differences on shrimps. Among these, we chose the triangular test due to its accurate answer (forced choice) from the tasters’ experience (UNI EN ISO 13299:2016). For each storage phase and each taster, three differently coded samples were presented, two of which were identical and one different, developing every possible combination between the types of treatment (Table 3).

Results at the time D0 showed significant differences (α = 0.01) between PE and the S shrimps, PE vs. CTRL and S vs. CTRL, whereas significant (α = 0.01) similarities were observed between CTRL and PE+S and PE+S vs. S. This is likely due to the fact that both PE and S samples could be characterized by sensory properties given by the additives used, which allowed the tasters to widely differentiate one from the other and when they were compared to the CTRL. This hypothesis was corroborated by comparing PE+S and PE or S or CTRL, which resulted in not being significantly different from each other when we used halved concentrations of sulphite and phenolic extract in the PE+S shrimps.

After 3 and 6 days of storage (D3 and D6), significant differences (α = 0.01) remained between CTRL and PE and CTRL and S shrimps (α = 0.01).

After a further 8 days of storage (D8), significant differences (α = 0.01) emerged not only between the CTRL and S shrimps and CTRL and PE shrimps, which appeared sensory different during storage, but also between S vs. PE samples.

The results indicate that the use of the phenolic extract and the sulphite additive deeply modified the overall sensory perception of the shrimps since CTRL vs. S and CTRL vs. PE were significantly different from each other.

Furthermore, these significant differences were also probably due to the strong sensory modifications undergone by CTRL shrimps, which were unprotected by any type of preservatives. Thus, the microbial proliferation contributed to significantly modify the original overall sensory quality, already demonstrated by other authors [6,7]. In fact, the only triangle test which involved CTRL shrimps resulting in not being significantly different, was between the CTRL and PE+S shrimps at time D6, but results showed an equal distribution (18 vs. 18) between odd and right answers.

These results led to the conclusion that significant sensory changes occurred during shrimp storage, both as a consequence of the use of phenolic extract and sodium metabisulfite additives, and of storage time. However, since sensory discriminant methods, such as the triangle test, did not give the possibility to describe these sensory changes, in order to have a qualitative-quantitative evaluation of the evolution of the sensory quality of shrimp samples during storage and the sensory impact of the additives used, an analytical descriptive sensory analysis was performed.

#### QDA Test

An analytical assessment of the evolution of the sensory profile of the three shrimps’ typologies over time was carried out through the application of QDA, following the UNI EN ISO 13299 method (2016). Indeed, through the development of an unstructured quantitative descriptive analytical sheet, it was possible to obtain the sensory profile of the shrimps subjected to the different treatments and the different storage times. In this regard, a form has been adopted including descriptors both according to Erickson et al. [23] and preliminary sessions which included the evaluation of CTRL, S, PE, and PE+S shrimps at D0, allowing to complete the profile sheet with the following descriptors: Red/orange colour, brown colour, presence of spots, paste colour homogeneity, gloss, for the appearance sensation; overall odour, ocean/sea water, cooked shrimps, old shrimps, and brine, for the olfactory sensation; sweet, salty, sour, bitter, brine, and overall taste intensity for the taste sensation; firmness, juiciness, chewability, crunchiness, fibrousness, for the texture and mouthfeel sensations, astringency, burning, metallic; for the tactile sensations; aftertaste, aftertaste of iodine, persistence, for the final sensations. However, assessors were free to fill the profile sheet with descriptors which were not included in the sheet and eventually emerged by the shrimps quality evaluation overtime, through the section ‘others’. In this regard, assessors evaluated the typical bitter and pungent sensations of the olive phenolic compounds [40] as, not perceptible in both PE and PE+S shrimps, at the levels used in this experimentation (2 and 1 g/L, respectively). However, the bitter descriptor was included to describe a negative taste sensation due to the shrimps’ flesh deterioration. Furthermore, at those PE concentrations, no specific odour correlated to the olive or olive vegetation water, was revealed in PE and PE+S shrimps, highlighting results in contrast to other research, where the use of natural additives for seafood preservation seems limited by the high negative sensory impact of these [41,42].

The collected sensory data were used for the multivariate statistical elaboration. An exploratory PCA model was built to describe the dispositions of objects (shrimp) in a multidimensional space according to the treatment and evolution of all parameters, during storage for 8 days (Figure 3), reducing the size of data and eliminating the redundant data. The PCA of the first three components (68% explained variance) showed clustering based on days of storage in the first component (left to right side of the score plot) and broad discrimination according to the treatment that occurred to the shrimp along the second component (bottom to top side of the score plot) (Figure 3a). The relative loading plot (Figure 3b) shows that the variables most responsible for the distribution of the fresher samples and hence, the most responsible for their left location in the score plot, were antioxidant activity hydroxytirosol, and the sensory descriptors such as ‘sweet, ‘juicy, ‘overall odour intensity’, ‘flesh colour homogeneity’. On the contrary, the variables characterized by the highest loading, responsible for the right-hand location on the score plot of the older shrimps were *Enterobacteriaceae* count, L* and b* parameters, and the following sensory descriptors: ‘Dried shrimps’, ‘ammonia odour’, ‘ammonia taste’, ‘sour’, and ‘bitter’. Thus, the PCA allowed us to confirm that the longer the time of the shrimps storage, the higher the concentration of *Enterobactriaceae*, as well as negative sensory attributes correlated to the ammonia taste an odour. These are typical off-flavours arising from fish flesh protein deterioration which promote N compounds accumulation responsible for the fish off-flavours [43]. Looking at the second component of the PCA (from the bottom to the top of the score plot; Figure 3a), a wide separation of the samples according to the preservation treatment was evident with the CTRL and the S samples located in the lower part of the plot, without a clear differentiation between the two typologies and the other two groups, represented by PE and PE+S samples, at the opposite end. Specifically, PE+S shrimps occupied the intermediate zone, whereas the PE shrimps were located on all of the other three groups. The relative loading plot (Figure 3b) showed that PE shrimps (located in the upper part of the score plot) were characterised by the highest values of verbascoside, tyrosol, sum of the penolic fractions, as well as by the sensory attributes ‘crunchiness’, ‘brightness’, ‘fibrousness’, and ‘spiciness’, confirming that PE shrimps were characterised by the presence of secoridoid derivatives in their flesh over time, but also by high scores of positive sensory attributes. Thus, PCA allowed us to bring into focus the fact that the recognized phenolic antioxidant and antimicrobial activities [17,40,44], allowed the shrimp to limit the accumulation of sensory defects and prolong their freshness characteristics.

In contrast, the highest loading variables for the CTRL and S samples (located in the lower part of the score plot, Figure 3a), were counts of higher values of psychrotrophic and mesophilic bacteria as well as ‘brine’ and ‘bitter’ taste, different from the lower values of phenolic fraction and positive sensory attributes characterizing the PE group. Furthermore, PE+S shrimps, occupying the central part of the score plot (Figure 3a), were characterised by intermediated values of all the variables cited above (Figure 3b). This highlights that both CTRL and S samples were characterised by the same variables overtime, assuming a similar loss of quality during shelf life, thus lower antimicrobial and antioxidant activities of the sulphite additive compared to the phenolic extract.

To confirm the qualitative differences among the samples according to the storage time and treatments, highlighted by PCA results, and to assess correlations between all the quality parameters and treatment types, an orthogonal partial least squares discriminant analysis (O-PLS-DA) was employed (58% total explained variance with a predictive component and an orthogonal component which explained 19% and 39%, respectively) (Figure 4). The score plot (Figure 4a) which projected the components T0 [1] vs. T1 showed a wider separation along the principal predictive component (from the left to the right side of the score), according to the shrimps’ treatment, as observed in the PCA score. In particular, we found PE samples located on the left of the score, CTRL and S on the opposite side, and the shrimps of the PE+S group in the middle of the score (Figure 4a). Furthermore, along the orthogonal component, (T [1], from the lower to the upper side of the score plot, Figure 4a), the objects (shrimps) separation was related to the time of storage with the D0 and D3 samples and D6 and D8 samples located in the lower and in the upper part of the score plot, respectively. In the relative Pq [1] vs. poso [1] loading plot (Figure 4b), we found high correlations between the independent variables verbascoside, tyrosol, hydroxytyrosol, the sum of the phenolic fraction and the antioxidant activity and the latent dependent variable represented by the PE shrimps (Figure 4b, left side). This confirmed the ability of the phenolic substances contained in the PE, to maintain high antioxidant activity of the shrimps’ cooked flesh as other researches have already demonstrated [22,45] during heating processes. Furthermore, the loading plot also shows high correlation between the latent variable representing PE shrimps and many of the sensory positive attributes such as crunchiness, brightness, sweetness, and juiciness (Figure 4b, left side). In contrast, the latent dependent variables S and CTRL samples showed low correlation with all the variables characterizing PE samples and high correlation with psychrotrophic and mesophilic bacteria and Enterobacteriaceae, that were negatively correlated with raw PE samples (Figure 4b, right side). Indeed, the presence of polyphenols in PE shrimps helped limit bacterial proliferation, confirming previous results [17].

On the other hand, cooked S and CTRL shrimps, showed high correlation with some negative sensory descriptors such as bitter, sour, and ammonia taste, which generally are associated with the decomposition phenomena of the fish proteins carried out by bacteria [46] (Figure 4b, right side). In this regard, both QDA and multivariate statistical analyses were very useful to better investigate the reasons which determine the sensory quality differences among samples. In fact, while the triangle test revealed significant differences between CTRL vs. S and CTRL vs. PE shrimps overtime, all the data collected and elaborated by PCA and O-PLS-DA showed that CTRL and S were very similar in terms of overall quality evolution. On the contrary, significant differences were confirmed between CTRL and PE and S and PE samples. In addition to proving the strong effect of the antioxidant capacity, these statistical analyses revealed the higher capacity of PE in preserving the overall quality of the shrimps during the storage period, rather than sodium metabisulphite. This is likely due to the fact that these substances protected the shrimps’ flesh against oxidation and microbial degradation, and the same is reflected in the variables associated with this process.

## 4. Conclusions

The results presented in this paper provide evidence on the effectiveness of the phenolic extract from a waste product, such as olive milling vegetation water, both from a hygienic point of view and in terms of shrimp quality in general. Interpretation of the sensory test using an exploratory PCA model and correlation analyses revealed significant improvements conferred by the extract to the microbiological profile and consequently to the sensory characteristics during storage. The bactericidal and antioxidant activity of the phenolic compounds was proportional to their concentration, as were the pleasant sensory attributes. In this regard, the phenolic fractions found in the shrimps after cooking, albeit at different percentages of reduction, enriched the nutraceutical value of the rose shrimps, which in addition to being safer with longer-lasting shrimp sensory characteristics, were also without or with less use of synthetic additives. Furthermore, at the concentrations used by the authors, the phenolic extract did not modify either the sensory characteristics of the fresh shrimp or during the storage period. Finally, sulphites combined with the extract demonstrated a less effective antibacterial action, although they appear to have protected the phenolic compounds from the oxidative processes of cooking and storage. Therefore, this strategy could be a valid alternative that would make the total or partial reduction of the concentration of synthetic additives in the food sector possible.

## Figures and Tables

**Figure 1 foods-10-02116-f001:**
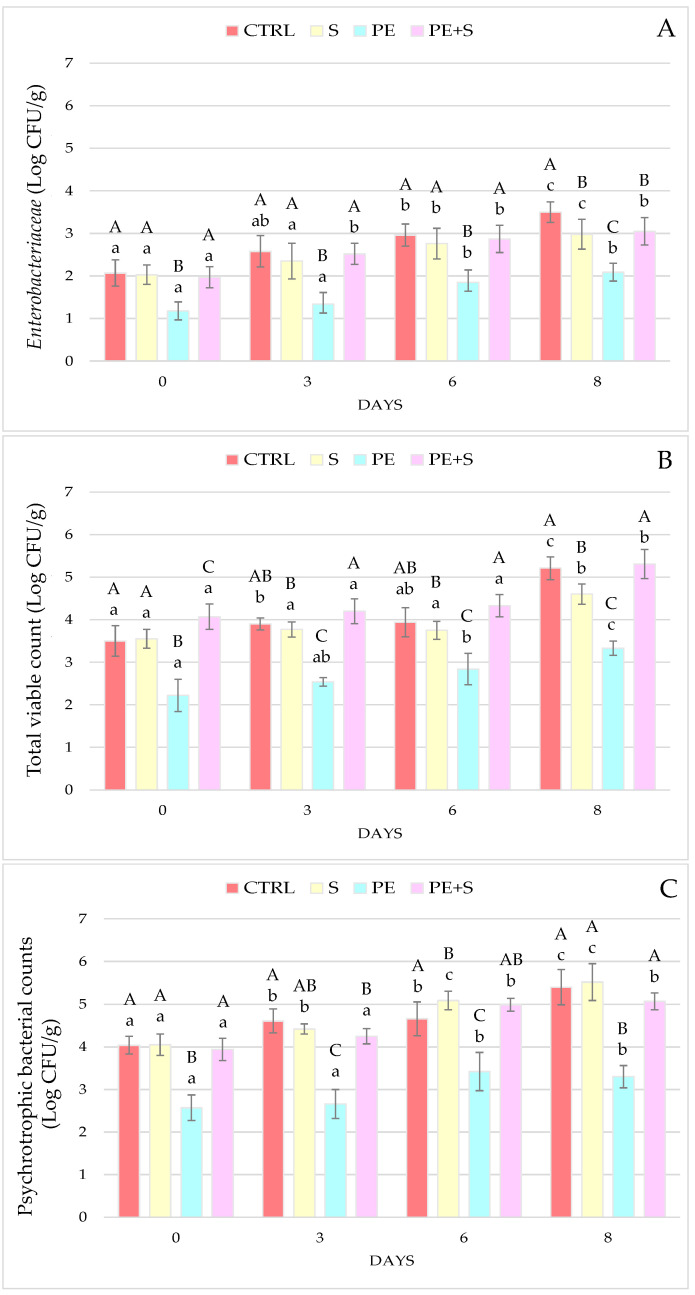
Microbiological changes at days 0, 3, 6, and 8 of storage, in control (CTRL), sulphites (S), phenolic extract (PE), and phenolic extract + sulphites (PE+S) shrimps. Values are means ± standard deviation. Different uppercase letters, within each day of storage, represent significant differences between groups (*p* < 0.05); different lowercase letters, within each group, represent significant differences between days of storage (*p* < 0.05). *Enterobacteriaceae* (**A**), total viable count (**B**), and psychrotrophic bacteria counts (**C**).

**Figure 2 foods-10-02116-f002:**
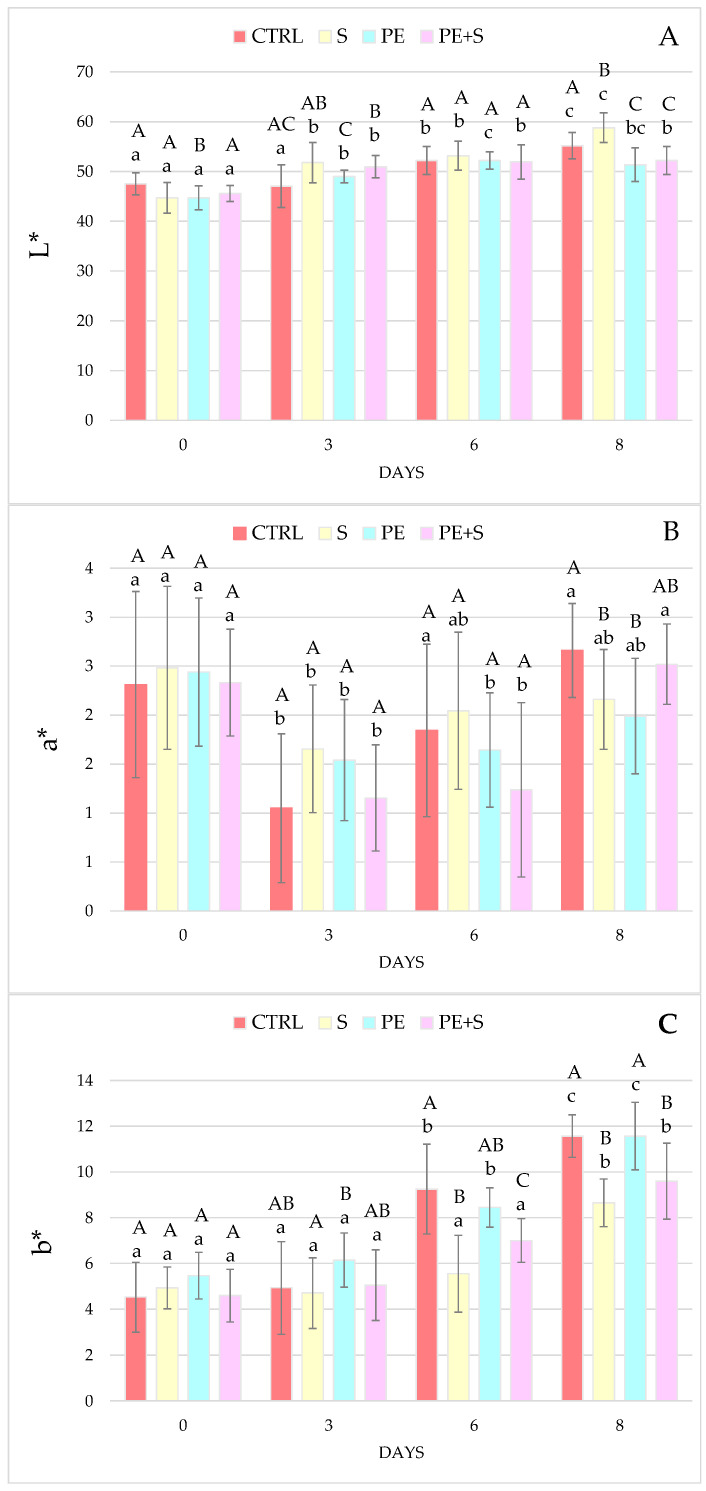
Color changes at days 0, 3, 6, and 8 of storage, in control (CTRL), sulphites (S), phenolic extract (PE), and phenolic extract + sulphites (PE+S) shrimps. Values are means ± standard deviation. Different uppercase letters, within each day of storage, represent significant differences between groups (*p* < 0.05); different lowercase letters, within each group, represent significant differences between days of storage (*p* < 0.05). L* (**A**), a* (**B**), and b* (**C**).

**Figure 3 foods-10-02116-f003:**
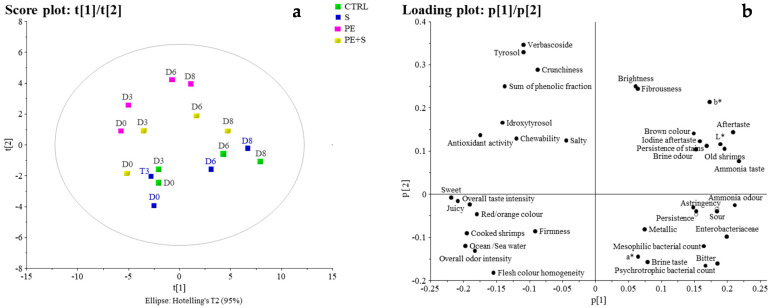
Score (**a**) and loading plot (**b**) of the PCA model built with all the samples and variables. The model with three significant principal components explains 68% of the variance of the data (46%, 13%, and 9%).

**Figure 4 foods-10-02116-f004:**
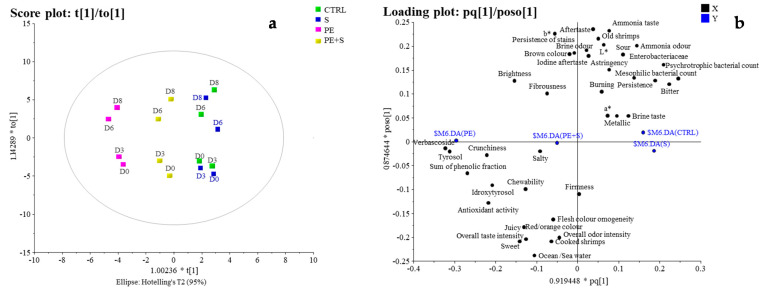
Score (**a**) and loading plot (**b**) of the OPLS-DA model built with all the samples and the variables. The model with one predictive component and one orthogonal component explains 58% of the variance of the data (19% and 39%, respectively).

**Table 1 foods-10-02116-t001:** Phenolic compounds (mg/kg) evolution during storage in raw shrimps (before boiling) treated with tap water solution containing 2 g/L of phenols (PE) and 0.25% sodium metabisulfite tap water solution containing 1 g/L of phenols (PE+S).

		3,4-DHPEA	p-HPEA	Verbascoside	3,4-DHPEA-EDA	Sum of Polyphenols
Days	Group	Mean ± SD	Mean ± SD	Mean ± SD	Mean ± SD	Mean ± SD
0	PE	250.18 ± 0.44	51.28 ± 0.17	81.44 ± 0.48	227.36 ± 4.00	610.27 ± 5.08
	PE+S	211.24 ± 0.82	30.70 ± 0.32	33.11 ± 0.95	185.67 ± 2.01	460.72 ± 4.10
3	PE	306.52 ± 0.15	49.83 ± 0.23	80.32 ± 0.64	78.48 ± 1.93	515.15 ± 2.95
	PE+S	221.35 ± 0.20	28.36 ± 0.17	31.82 ± 0.88	127.51 ± 1.21	409.05 ± 2.46
6	PE	229.60 ± 0.14	49.77 ± 0.51	76.23 ± 0.27	20.14 ± 0.31	375.74 ± 1.23
	PE+S	210.55 ± 0.40	31.90 ± 0.21	33.77 ± 0.42	66.88 ± 0.73	343.09 ± 1.81
8	PE	148.20 ± 0.26	48.14 ± 0.19	75.13 ± 1.01	0.00 ± 0.00	271.47 ± 1.46
	PE+S	205.92 ± 0.52	27.33 ± 0.28	30.36 ± 0.26	0.00 ± 0.00	263.61 ± 1.06

SD: standard deviation.

**Table 2 foods-10-02116-t002:** Phenolic compounds (mg/kg) evolution during storage in cooked shrimps (after boiling) treated with tap water solution containing 2 g/L of phenols (PE) and 0.25% sodium metabisulfite tap water solution containing 1 g/L of phenols (PE+S).

		3,4-DHPEA	p-HPEA	Verbascoside	3,4-DHPEA-EDA	Sum of Polyphenols
Days	Group	Mean ± SD	Mean ± SD	Mean ± SD	Mean ± SD	Mean ± SD
0	PE	109.50 ± 0.21	35.50 ± 0.36	45.80 ± 0.32	0.00 ± 0.00	190.80 ± 0.89
	PE+S	120.03 ± 0.31	25.70 ± 0.22	18.80 ± 0.03	0.00 ± 0.00	164.53 ± 0.55
3	PE	93.39 ± 0.61	29.82 ± 0.46	48.13 ± 0.85	0.00 ± 0.00	171.34 ± 1.92
	PE+S	103.67 ± 0.28	21.68 ± 0.46	17.42 ± 0.09	0.00 ± 0.00	142.77 ± 0.82
6	PE	64.74 ± 1.03	28.53 ± 0.11	40.05 ± 0.92	0.00 ± 0.00	133.32 ± 2.06
	PE+S	98.04 ± 0.13	24.93 ± 0.06	20.87 ± 0.03	0.00 ± 0.00	143.84 ± 0.22
8	PE	0.00 ± 0.00	27.77 ± 0.00	34.27 ± 1.18	0.00 ± 0.00	62.03 ± 1.78
	PE+S	0.00 ± 0.00	18.44 ± 0.16	16.56 ± 0.62	0.00 ± 0.00	35.00 ± 0.78

SD: standard deviation.

**Table 3 foods-10-02116-t003:** Results of the sensory discriminant triangle test assessed in control (CTRL), sulphites (S), phenolic extract (PE), and phenolic extract + sulphites (PE+S) shrimps at days 0, 3, 6, and 8 of storage.

Days of Storage	Comparison between Samples	No. of Total Answers	No. of Correct Answers	No. of Wrong Answers	Significant Difference between Samples *
0	CTRL vs. PE	36	22	14	YES
	CTRL vs. PE+S	36	21	15	NO
	CTRL vs. S	36	22	14	YES
	PE vs. S	36	30	6	YES
	PE vs. PE+S	36	18	18	NO
	PE+S vs. S	36	12	18	NO
3	CTRL vs. PE	36	30	6	YES
	CTRL vs. PE+S	36	21	15	NO
	CTRL vs. S	36	36	0	YES
	PE vs. S	36	20	16	NO
	PE vs. PE+S	36	19	17	NO
	PE+S vs. S	36	12	24	NO
6	CTRL vs. PE	36	30	6	YES
	CTRL vs. PE+S	36	18	18	NO
	CTRL vs. S	36	30	6	YES
	PE vs. S	36	20	16	NO
	PE vs. PE+S	36	21	15	NO
	PE+S vs. S	36	18	18	NO
8	CTRL vs. PE	36	30	6	YES
	CTRL vs. PE+S	36	22	14	YES
	CTRL vs. S	36	30	6	YES
	PE vs. S	36	30	6	YES
	PE vs. PE+S	36	18	18	NO
	PE+S vs. S	36	21	15	NO

* The significance was evaluated according to the values previously chosen for α = 0.001 and Pd = 50%.

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
