# Peer review of "Quality Evaluation of Shrimp (Parapenaeus longirostris) Treated with Phenolic Extract from Olive Vegetation Water during Shelf-Life, before and after Cooking"

_foods, 2021, doi:10.3390/foods10092116_

Round 1

Reviewer 1 Report

The manuscript entitled "Quality evaluation of cooked shrimp (Parapenaeus longirostris) treated with phenolic extract from olive vegetation water during shelf-life" is interesting and well designed, however it should have more trials to evaluate the effects of incorporation of the phenolic fraction, namely trials of toxicity.
The statistical treatment of the results is very interesting but the discussion of the results obtained can be improved.

Author Response

Please, see the attached file 

Reviewer 2 Report

A well-written and interesting paper. However, some points should be revised. Please verify:
Line 49: "harvest" is this the adequate word?

Line 131- 133. The "cooking procedure" should be described in section 2.1 Experimental design methodology.

Colour evaluation is not described in the Material and Method section.

Please present phenolic data in a figure or table form.  

Author Response

Please, see the attached file

Reviewer 3 Report

I am convinced that the manuscript by Miraglia and co-workers provides lots of valuable data.

Remarks, queries and suggestions:

  • Title: there is about cooked shrimps but some data (microbiological for instance) are related to not cooked samples. Any idea to change the title?
  • Description of the PE+S treatment. It should be cleared what was the thought behind the combination of sodium metabisulfite and phenolic extract with specific dosages; I suppose that it was a half of dosage from group S and a half of the dosage taken from the PE one. Right?
  • Please make it clear for me and the readers that there is no similarities about the results published (for instance for microbiological results) in this paper and in the work of Miraglia et al. 2020 Foods, 9, 1647. The experimental schema seems to be the same.

I look forward to seeing your response.

Author Response

Please, see the attached file 

Round 2

Reviewer 2 Report

Table 1 and Table 2. Please clarify in the text and in the table legends the differences between these two tables.

Author Response

Differences between Table 1 and table 2 were clarified both in the legends and in the text.

Furthermore, for improving the English language and style, the paper has been carefully revised by a native English speaker.

Reviewer 3 Report

I accept the revision, Thank you.

Author Response

We thank the reviewer  3 or his/her appreciation of the quality of our work